# Exploring the intersections of sexual stigma, poverty and mental health in HIV-negative gay, bisexual and other men who have sex with men in the United States

Udodirim N. Onwubiko[1]*, Sarah M. Murray[2], Amrita Rao[3], Allison T. Chamberlain[1], Travis H. Sanchez[1], David Benkeser[4], David P. Holland[5,6], Samuel M. Jenness[1], Stefan D. Baral[3]

1 Department of Epidemiology, Rollins School of Public Health, Emory University, Atlanta, Georgia, United States of America, 2 Department of Mental Health, Johns Hopkins Bloomberg School of Public Health, Baltimore, Maryland, United States of America, 3 Department of Epidemiology, Center for Public Health and Human Rights, Johns Hopkins University, Baltimore, Maryland, United States of America, 4 Department of Biostatistics and Bioinformatics, Rollins School of Public Health, Emory University, Atlanta, Georgia, United States of America, 5 Fulton County Board of Health, Atlanta, Georgia, United States of America, 6 Division of Infectious Disease, School of Medicine, Emory University, Atlanta, Georgia, United States of America

* udodirim.onwubiko@emory.edu

## Abstract

Stigma related to non-heteronormative behavior remains a major challenge associated with mental health disparities among gay, bisexual, and other men who have sex with men (GBM). Economic hardship worsens these challenges, and characterizing these interactions can help inform effective mental health interventions for GBM. Using 2018 and 2019 American Men's Internet Survey data, we assessed population heterogeneity in sexual stigma experiences among adult, HIV-negative GBM using latent class analysis. We estimated associations between stigma patterns and mental health outcomes (psychological distress, suicidal ideation, and suicide attempt) using modified Poisson regression, quantifying the interaction between sexual stigma and poverty on multiplicative and additive scales. Four distinct sexual stigma patterns were identified that grouped GBM as experiencing: diverse forms of sexual stigma across multiple settings (12%); primarily anticipated stigma in healthcare settings (13%); predominantly enacted and perceived sexual stigma in family and general social settings (34%); or minimal sexual stigma (41%). Vulnerabilities to distinct stigma patterns varied by key participant demographics including age, nativity and education. Notably, the group with diverse stigma, particularly in the context of poverty, had significantly higher prevalence of serious psychological distress (aPR: 4.7 [95% CI: 3.9, 5.7]) and suicide attempts (aPR: 11.3 [95% CI: 6.6, 19.4]) compared to the group with minimal stigma and adequate income. These findings highlight the pivotal role of poverty in intensifying the impact of sexual stigma on the mental well-being of GBM. Addressing stigma within the broader context of structural determinants, including poverty, is crucial for optimizing mental health among GBM.

**Data Availability Statement:** The datasets analyzed in this study are available upon reasonable request, subject to approval by the Emory PRISM group. Requests can be directed to Travis Sanchez, PhD at travis.sanchez@emory.edu or submitted via the Emory AMIS website https://emoryamis.org/data-requests/.

**Funding:** This work was supported by the National Institutes of Health in the form of grants (R01MH128130 to SMJ; R01MH132150 and R01NR020437 to SDB; P30MH136919 to THS and SDB). The funders had no role in study design, data collection and analysis, decision to publish, or preparation of the manuscript.

**Competing interests:** The authors have declared that no competing interests exist.

## Background

Gay, bisexual, and other men who have sex with men (GBM) have been pivotal in advancing the HIV/AIDS response [1] in the United States (US). Despite efforts to reduce discrimination [2–5], GBM continue to face stigma associated with sexual behavior, which profoundly affects mental health. This persistent challenge underscores the need for a deeper understanding of stigma as a multifaceted concept. Stigma involves distinguishing and labeling differences, associating those labels with negative stereotypes, segregating individuals into 'them' versus 'us' categories, and ultimately subjecting labeled individuals to discrimination and status loss, resulting in unequal and adverse outcomes for affected populations [6]. Stigma operates at institutional, interpersonal, and individual levels. At the individual-level, sexual stigma manifests in various forms: as enacted or perceived stigma, where individuals experience direct or subtle discriminations based on their sexual identity; anticipated stigma, where individuals expect to face stigma in certain situations; and internalized stigma, where individuals accept a devalued identity as a part of his own value system and self-concept [7–10].

The link between sexual stigma and mental health is well supported by conceptual frameworks like Meyer's Minority Stress Theory (MST) and Hatzenbuehler's Psychological Mediation Framework (PMF) [11,12]. MST posits that stigma-based discriminations act as "distal" stressors, triggering psychological processes that generate "proximal" stressors–such as rejection expectation, stigma internalization, and identity concealment–which in turn, exacerbate psychological distress and increase the risk of mental health problems like mood disorders, substance use, and suicide-related outcomes [11]. Hatzenbuehler's PMF expands on this by proposing that the pathway between stigma-related stress and the onset of psychopathology is mediated by factors like heightened emotion dysregulation, increased social isolation, and cognitive difficulties [12].

Empirical evidence supports these theoretical models, showing that sexual minorities face elevated rates of harassment, assaults, property violence, and other manifestations of sexual stigma [13–16]. Research also reveals a strong link between internalized homophobia and suicide outcomes, often compounded by heightened emotion dysregulation [11,12,17–22]. For instance, a longitudinal study of over 1,000 students found that adolescents with same-sex attraction exhibited greater emotional dysregulation, such as heightened rumination and reduced emotional awareness, compared to their heterosexual peers [19]. Similarly, another study showed that heightened rumination mediated the relationship between internalized stigma and depression [18]. Overall, the literature consistently shows that sexual minorities face an earlier onset and higher prevalence of mood disorders, substance use, and both suicidal ideation and attempts [23,24] compared to their heterosexual peers [8,11,12,25].

Socioeconomic disparities present another significant challenge affecting health outcomes among GBM, exacerbated by systemic forces such as racism and heterosexism [26]. Discriminatory employment practices, wage gaps, and housing instability contribute to economic insecurity, which in turn leads to heightened stress, increased anxiety, a greater risk of stigma-related violence, and a higher likelihood of engaging in high-risk behaviors like exchange sex [27–29]. These socioeconomic challenges also hinder access to essential healthcare services, including HIV testing, preventive care, and mental health support, further complicating GBM'S ability to maintain overall well-being [28,30].

Several studies have examined how stigma and poverty influence mental health outcomes [31–34]. While much research has highlighted sexual stigma as a major factor driving higher rates of depression and other mental health issues among sexual minorities [31,33,35] others have investigated socioeconomic disadvantage (e.g., education, income, occupation), consistently linking lower socioeconomic status to poorer mental health [31,32]. However, evidence

indicates that sexual stigma and economic hardship are interrelated phenomena. Stigma can drive economic challenges through structural pathways that limit the resources available to stigmatized individuals, while poverty may amplify the negative effects of stigma by further restricting access to vital support resources [27,28,30]. Despite this intersection, few studies have explored the combined impact of sexual stigma and poverty on the mental health of HIV-negative US GBM, a group that may lack the enhanced mental health services available to those living with HIV [36].

This study aims to contribute to this gap by assessing patterns of sexual stigma experiences among HIV-negative GBM and exploring how those experiences relate to mental health outcomes within the context of economic hardship. We hypothesize that the combination of sexual stigma and economic hardship amplifies poor mental health outcomes. By investigating this interaction, we seek to better understand the broader impact of these social determinants on mental well-being and to inform evidence-based strategies to address mental health disparities among HIV-negative GBM.

## Methods

### Data source and study sample

Data from the 2018 and 2019 cycles of the American Men's Internet Survey (AMIS), an annual cross-sectional behavioral health survey for GBM, was used [37]. AMIS annually reaches over 10,000 MSM who are recruited through banner advertisements on social media, geospatial networking apps, email blasts and various gay interest websites. Detailed information on AMIS eligibility, recruitment and enrollment processes have been previously described [38,39]. Data collection for AMIS 2018 occurred between September and December, while data for AMIS 2019 were gathered from August through December. Although AMIS is primarily cross-sectional, a small fraction ($\leq 10\%$) of participants may have participated in previous survey years [40]. For this analysis, participants were included if they were adult (age $\geq 18$ years) cisgender man who reported oral or anal sex with another man in the past year, resided in the US, and reported being HIV-negative at the last screening.

### Ethics statement

Participants provided online informed consent before completing the survey, and no incentives were offered. All procedures involving human participants adhered to the ethical standards of Emory University's Institutional Review Board.

### Measures

Three mental health outcomes were assessed: psychological distress, suicidal ideation, and suicide attempts. Psychological distress was measured using the 6-item Kessler (K-6) scale, evaluating feelings of nervousness, hopelessness, restlessness, depression, general fatigue, and worthlessness over the past 30 days [41]. Responses were scored on a 5-point scale, with higher scores signaling more frequent feelings of distress. A total score of $\geq 13$ suggests clinically significant mental health distress and has been shown to strongly correlate with inpatient and emergency mental health service usage [42–45]. Using this cutoff, a binary measure, serious psychological distress, identified respondents for whom clinically significant mental distress was probable. The validity of K-6 as a screening tool for detecting clinically significant mental disorders from non-cases in US adult cohorts has been evaluated previously and found to be high [41,42]. It has also been shown to have high internal consistency, stable reliability, and dependable measurement invariance across various population age groupings [43,44]. Suicidal

ideation (or SI) was measured by asking "At any time in the past 12 months, up to and including today, did you seriously think about trying to kill yourself". Suicide attempt (or SA) was measured by asking "During the past 12 months, did you try to kill yourself?".

Fifteen AMIS survey items (**Table A in S2 Text**) were used to measure individual-level sexual stigma experiences across various contexts, all of which have been included in AMIS surveys since 2017. Respondents indicated presence and recency of these experiences, selecting from options: "Never", "Yes, but not in the past 6 months", and "Yes, in the past 6 months". To aid interpretability of latent class solutions (described below), the two "yes" responses were collapsed to create dichotomous stigma indicators that distinguished those with lifetime experiences from those without. Three indicators measured anticipated sexual stigma in healthcare and social settings, while four measured perceived stigmas in family, healthcare, social, and policing/law enforcement settings. The remaining eight indicators measured enacted sexual stigma across family, healthcare, and social settings. Among these, four indicators related to physical and sexual assault experiences were combined to create two indicators specifically identifying individuals whose experiences were linked to their sexual behavior. For example, respondents were coded as endorsing the new physical assault indicator only if they answered affirmatively to both the general question about physical assault ("Has someone ever physically hurt you (pushed, shoved, slapped, hit, kicked, choked, or otherwise physically hurt you)?") and a follow-up question linking the experience to their sexual behavior ("Do you believe any of these experiences were related to the fact that you have sex with men?").

Poverty was assessed using the income-to-need ratio (INR), which was derived from the annual pre-tax household income, household size, number of dependents younger than 18-years old reported by AMIS respondents, and the US Census Bureau's federal poverty thresholds (FPT) [46]. Household income data was collected using prespecified categories ($0-$19,999, $20,000-$39,999, $40,000-$74,999, $75,000 or more). To calculate the INR, a continuous measure of household income was approximated by assigning the midpoint of each reported category, with an assumed upper bound of $110,000 for the highest income category [47]. While this income assignment approach may truncate the INR distribution, it has been shown to outperform alternative methods of approximating continuous income from predefined categories, particularly in accurately reflecting actual income among individuals in lower income categories [47]. Calculated as the ratio of household income to designated FPT given survey year, respondent age and family composition, INR was used to dichotomize economic hardship among the study participants, categorizing those with INR < 2 as income-poor. This cutoff was selected to address the limitations of FPT as a measure which often fails to accurately capture the actual income needed for essential living expenses and to encompass a broader GBM population facing financial difficulties [48,49].

Additional factors considered were age, race/ethnicity (marker of cultural identity and potential structural racism experience), past-year homelessness, nativity (US-born versus foreign-born), US region of residence, and urbanicity determined by self-reported ZIP codes [50]. Urbanicity was categorized into four levels: large urban, large suburban, small/medium metropolitan, and rural (micropolitan and noncore) [50].

## Statistical analyses

Latent class analysis (LCA) was used to identify patterns of sexual stigma experiences within the study sample [51–54]. LCA, a model-based clustering methodology, uses posterior probabilities of class assignment based on maximum likelihood estimation with robust standard errors to detect unobserved population heterogeneity from responses to observed indicators [52,53,55]. Models with varying number of latent classes (ranging from 2 to 7) were fitted.

Optimal model selection was based on several criteria, including the Akaike Information Criterion (AIC), Bayesian Information Criterion (BIC) & sample-size-adjusted BIC (SABIC), modified likelihood ratio tests [52–55], theoretical interpretability of identified classes, and model entropy [52–56]. To assess measurement consistency across the two survey years, a three-step process was applied: first, the optimal number of best-fitting latent classes was evaluated for invariance; second, class-specific conditional response probabilities were assessed for consistency; and third, class prevalence was examined [57,58]. Differences in the optimal number of classes or degradation of model fit in fully or partially constrained models indicated a lack of consistency (see **S1 Text**).

After selecting the optimal latent class model, participants were assigned to stigma classes using estimated posterior probabilities of membership [52,53]. Associations between stigma classes and participant characteristics were assessed using the bias-adjusted, 3-step multinomial logistic regression approach in Mplus [53,59,60].

Modified Poisson regression was then used to evaluate the relationship between the identified classes and mental health outcomes, with confounder selection guided by a directed acyclic graph (**Fig A in S2 Text**). interaction between stigma classes and poverty was explored by incorporating cross-product terms and was quantified on both multiplicative and additive scales [61]. Potential misclassification bias from stigma class uncertainty was addressed through record-level probabilistic bias adjustment [62,63]. Further methodological details are provided in the **S1 Text**.

Except for poverty measures which had non-response rates up to 25%, overall missingness was low ($\leq$10%), and multiple imputation was used to assess the impact of missing data on estimated associations. Sensitivity analysis was conducted to explore the effects of using recent versus lifetime stigma experiences and varying INR cutoff points on the associations.

All P values were two-sided with a significance level of 0.05. LCA was performed using Mplus (version 8), while all other analyses were conducted in R (v4.2.0) [64,65].

## Results

### Sample description & sexual stigma experience endorsement

Of 12,500 eligible GBM, five were excluded due to non-response in all sexual stigma items. Of 12,495 GBM included in the analysis (**Table 1**), 51% (n = 6370) were under 30 years old, and 52% (n = 6476) held a 4-year college degree or higher. The majority were non-Hispanic white (68%, n = 8314), while non-Hispanic Black and Hispanic GBM comprised 9% (n = 1146) and 15% (n = 1850) of the sample, respectively. About 20% were categorized as income-poor. Few meaningful differences in participant characteristics were observed between survey cycles; the 2019 cycle had a slightly higher proportion of respondents under 30 (2018: 47% vs. 2019: 55%, p<0.001) and Black non-Hispanic participants (2018: 5% vs. 2019: 14%, p<0.001).

The most frequently endorsed sexual stigma experience was receiving discriminatory remarks from family members (52%, 5825/11237), while the least endorsed was being gossiped about by healthcare providers (4%, 523/ 12158). Stigma experience endorsements were consistent across survey cycles. Additional details on endorsement variations by age and race/ethnicity are presented in **Table B in S2 Text**.

### Latent sexual stigma classes

Information criteria, entropy and likelihood ratio tests estimated for all fitted LCA models are presented in **Table C in S2 Text**. While BIC and other information criteria continued to decrease with increasing classes, they exhibited a noticeable point of diminishing returns (an "elbow") at the 4-class solution (**Fig 1A**). Entropy in the 4-class solution also exceeded 0.8,

**Table 1. Demographic characteristics and sexual stigma item endorsement among HIV-negative men who have sex with men who responded to the American Men's Internet Survey (AMIS), AMIS 2018–2019.**

| Characteristics | Total n (column %) | AMIS 2018 n (column %) | AMIS 2019 n (column %) | p-value* |
|---|---|---|---|---|
| N | 12495 | 6017 | 6478 | |
| **Age (years):** Median (Q1, Q3) | 29 (23, 45) | 30 (23, 48) | 28 (23, 43) | < 0.001 |
| **Age Categories (years)** | | | | < 0.001 |
| 18–24 | 3952 (32%) | 1827 (30%) | 2125 (33%) | |
| 25–29 | 2418 (19%) | 1000 (17%) | 1418 (22%) | |
| 30–39 | 2266 (18%) | 1157 (19%) | 1109 (17%) | |
| 40 or older | 3859 (31%) | 2033 (34%) | 1826 (28%) | |
| **Race/Ethnicity** | | | | < 0.001 |
| Black, non-Hispanic | 1146 (9%) | 285 (5%) | 861 (14%) | |
| Hispanic | 1850 (15%) | 896 (15%) | 954 (15%) | |
| Other or multiple races | 972 (8%) | 442 (7%) | 530 (8%) | |
| White, non-Hispanic | 8314 (68%) | 4304 (73%) | 4010 (63%) | |
| **Country of Birth:** Foreign-Born | 940 (8%) | 469 (8%) | 471 (7%) | 0.267 |
| **Education** | | | | 0.230 |
| HS or less | 1584 (13%) | 731 (12%) | 853 (13%) | |
| Some college or technical training | 4380 (35%) | 2104 (35%) | 2276 (35%) | |
| College degree or postgraduate education | 6476 (52%) | 3144 (53%) | 3332 (52%) | |
| **Annual Household Income (Pre-Tax)** | | | | 0.023 |
| $0–20000 | 1566 (13%) | 727 (13%) | 839 (14%) | |
| $20000–40000 | 2408 (21%) | 1146 (20%) | 1262 (21%) | |
| $40000–75000 | 3465 (30%) | 1649 (29%) | 1816 (30%) | |
| $75000 or more | 4246 (36%) | 2132 (38%) | 2114 (35%) | |
| **Income-to-Need Ratio (INR):** Median (Q1, Q3) | 3.5 (2.3, 5.4) | 3.6 (2.3, 5.5) | 3.5 (2.3, 4.7) | 0.033 |
| **Income-poor (INR<2):** Yes | 1917 (20%) | 827 (20%) | 1090 (21%) | 0.733 |
| **Homeless in past 12 months:** Yes | 952 (8%) | 384 (7%) | 568 (9%) | 0.001 |
| **US Region** | | | | < 0.001 |
| Northeast | 2098 (17%) | 973 (16%) | 1125 (17%) | |
| Midwest | 2540 (20%) | 1257 (21%) | 1283 (20%) | |
| South | 4975 (40%) | 2285 (38%) | 2690 (42%) | |
| West | 2882 (23%) | 1502 (25%) | 1380 (21%) | |
| **Urbanicity** | | | | 0.013 |
| Large Urban | 4936 (40%) | 2383 (40%) | 2553 (39%) | |
| Large Suburban | 2536 (20%) | 1153 (19%) | 1383 (21%) | |
| Small/Medium Urban | 3922 (31%) | 1946 (32%) | 1976 (31%) | |
| Rural | 1098 (9%) | 534 (9%) | 564 (9%) | |
| **Sexual Stigma Item Endorsement** | n (%) [a] | n (%) [a] | n (%) [a] | p-value |
| **Have you ever _______ because you have sex with men?** | | | | |
| felt excluded from family activities (A1) | 4112 (35%) | 1959 (35%) | 2153 (35%) | 0.604 |
| felt that family members have made discriminatory remarks or gossiped about you (A2) | 5825 (52%) | 2715 (51%) | 3110 (53%) | 0.035 |
| felt rejected by your friends (A3) | 3352 (28%) | 1543 (27%) | 1809 (29%) | 0.015 |
| felt afraid to go to health care services because you worry someone may learn (B1) | 3341 (27%) | 1573 (26%) | 1768 (28%) | 0.168 |
| avoided going to health care services because you worry someone may learn (B2) | 2642 (21%) | 1235 (21%) | 1407 (22%) | 0.139 |
| heard health care providers gossiping about you (talking about you) (B3) | 523 (4%) | 256 (4%) | 267 (4%) | 0.608 |
| felt that you were not treated well in a health center because someone knew (B4) | 1158 (10%) | 559 (10%) | 599 (9%) | 0.791 |
| felt that the police refused to protect you (C1) | 948 (8%) | 463 (8%) | 485 (8%) | 0.583 |

*(Continued)*

**Table 1.** (Continued)

| Characteristics | Total<br>n (column %) | AMIS 2018<br>n (column %) | AMIS 2019<br>n (column %) | p-value* |
|---|---|---|---|---|
| felt scared to be in public places (C2) | 4820 (39%) | 2255 (38%) | 2565 (40%) | 0.03 |
| been verbally harassed and felt it was (C3) | 5794 (47%) | 2801 (48%) | 2993 (47%) | 0.447 |
| been blackmailed by someone (C4) | 1666 (14%) | 723 (12%) | 943 (15%) | < 0.001 |
| Has someone ever physically hurt you (pushed, shoved, slapped, hit, kicked, choked, or otherwise physically hurt you) (D1) | 2066 (17%) | 956 (16%) | 1110 (18%) | 0.059 |
| been forced to have sex when you did not want to (i.e., physically forced, coerced to have sex, or penetrated with an object, when you did not want to) (D3) | 1313 (11%) | 529 (10%) | 784 (13%) | < 0.001 |

*P-value–Tests for differences in distributions of participant characteristics by survey cycle (Wilcoxon rank sum test used for continuous variables and Chi-square tests for categorical variables).

n (%)[a]–number and percent of participants endorsing stigma experience.

signifying good separation between identified classes. Likelihood ratio tests indicated that the 4-class solution significantly improved data fit over the 3-class solution (p<0.0001).

Examination of conditional response probabilities (Fig 1B) estimated in the 4-class model revealed distinct patterns of stigma experiences, grouping GBM into: 1) those with diverse stigma experiences across multiple settings (group labeled as the "Diverse Sexual Stigma class"), 2) those with predominantly anticipated stigma in healthcare settings (labeled "the Anticipated Healthcare Predominant Sexual Stigma class"), 3) those with predominantly enacted and perceived sexual stigma experiences in family and general social settings ("the Family and General Social Sexual Stigma class"), and 4) those with generally minimal stigma

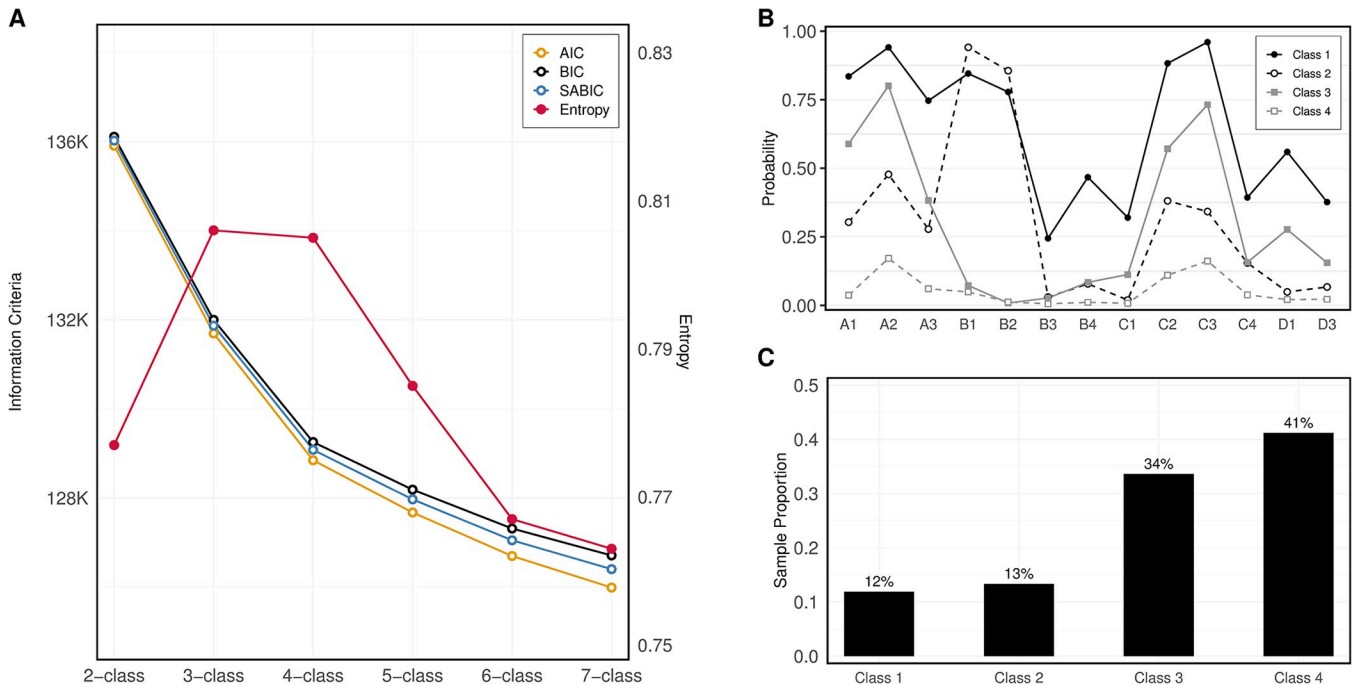

**Fig 1.** Sexual Stigma Measurement Among AMIS 2018–2019 Respondents: Parsimony Indices and Entropy for 2-7-class Maximum Likelihood Models (**A**), Conditional Response Probabilities (CRPs) for the 4-class Model (**B**) & Latent Sexual Stigma Class Prevalence in Study Sample (**C**).

experiences ("the Minimal Sexual Stigma class"). These groups constituted 12%, 13%, 34% and 41% of the sample, respectively (**Fig 1C**).

Stigma measurement invariance assessment showed consistent number of classes, conditional response probabilities, and class prevalences across both survey year data, confirming the stability of latent stigma measurement across the two cycles (**Table D in S2 Text**). Sensitivity analysis of stigma indicator dichotomization based on experience recency (**Table F in S2 Text**) also supported the 4-class model as optimal.

### Sociodemographic characteristics associated with stigma classes

Participants identified as experiencing diverse stigma or predominantly family and general social stigma were more likely to be younger, have faced homelessness in the past-year, live in poverty, and reside in the US South or Midwest compared to those in the minimal stigma category (**Table 2**). Those identified with predominantly anticipated healthcare sexual stigma were more likely to be younger, foreign-born (adjusted odds ratio [aOR] 1.43 [95% confidence

**Table 2. Sociodemographic characteristics of gay, bisexual and other men who have sex with men associated with latent sexual stigma class identification, AMIS 2018–2019.**

| | Diverse-SBS Class | | Anticipated Healthcare Predominant SBS Class | | Family and General Social Predominant SBS Class | |
|---|---|---|---|---|---|---|
| | Adjusted OR (95% CI) | p value | Adjusted OR (95% CI) | p value | Adjusted OR (95% CI) | p value |
| **Age (years)** | | | | | | |
| 18–24 | **1.51 (1.21, 1.89)** | **0** | **2.16 (1.74, 2.69)** | **0** | **1.63 (1.38, 1.93)** | **0** |
| 25–29 | **1.63 (1.29, 2.06)** | **0** | **1.81 (1.44, 2.27)** | **0** | **1.48 (1.25, 1.77)** | **0** |
| 30–39 | **1.40 (1.11, 1.77)** | **0.005** | **1.40 (1.11, 1.77)** | **0.004** | **1.15 (0.97, 1.38)** | **0.115** |
| 40 or older | *Ref* | | *Ref* | | *Ref* | |
| **Race/Ethnicity** | | | | | | |
| Black, non-Hispanic | **0.36 (0.26, 0.51)** | **0** | **0.72 (0.55, 0.94)** | **0.015** | **0.47 (0.37, 0.58)** | **0** |
| Hispanic | **0.63 (0.48, 0.81)** | **0** | 0.87 (0.69, 1.11) | 0.265 | **0.70 (0.58, 0.85)** | **0** |
| Other or multiple races | 0.88 (0.64, 1.20) | 0.417 | 0.81 (0.59, 1.12) | 0.195 | 0.96 (0.76, 1.22) | 0.766 |
| White, non-Hispanic | *Ref* | | *Ref* | | *Ref* | |
| **Education** | | | | | | |
| HS or less | **0.65 (0.49, 0.87)** | **0.004** | 0.75 (0.56, 1.01) | 0.057 | **0.76 (0.61, 0.93)** | **0.008** |
| Some college or technical training | *Ref* | | *Ref* | | *Ref* | |
| College degree or postgraduate education | 1.17 (0.97, 1.40) | 0.100 | **1.51 (1.25, 1.81)** | **0** | 1.04 (0.90, 1.19) | 0.627 |
| **Country of Birth:** Foreign-Born | 0.93 (0.65, 1.32) | 0.679 | **1.43 (1.08, 1.91)** | **0.013** | 1.00 (0.78, 1.28) | 0.971 |
| **Homeless in past 12 months:** Yes | **3.84 (2.94, 5.02)** | **0** | 0.94 (0.62, 1.42) | 0.778 | **2.04 (1.59, 2.61)** | **0** |
| **Income Poverty:** Income-poor | **1.66 (1.35, 2.04)** | **0** | **1.29 (1.05, 1.60)** | **0.018** | **1.39 (1.18, 1.64)** | **0** |
| **US Region** | | | | | | |
| Northeast | *Ref* | | *Ref* | | *Ref* | |
| Midwest | **1.33 (1.02, 1.72)** | **0.034** | **1.34 (1.02, 1.76)** | **0.033** | **1.25 (1.02, 1.53)** | **0.032** |
| South | **1.40 (1.10, 1.77)** | **0.006** | **1.62 (1.27, 2.05)** | **0** | **1.52 (1.27, 1.82)** | **0** |
| West | 1.09 (0.83, 1.44) | 0.525 | **1.52 (1.17, 1.98)** | **0.002** | **1.29 (1.06, 1.57)** | **0.012** |
| **Urbanicity** | | | | | | |
| Large Urban | *Ref* | | *Ref* | | *Ref* | |
| Large Suburban | 0.80 (0.63, 1.01) | 0.065 | 1.16 (0.94, 1.44) | 0.175 | 0.90 (0.76, 1.07) | 0.240 |
| Small/Medium Urban | 0.93 (0.77, 1.13) | 0.471 | 1.05 (0.87, 1.27) | 0.618 | **0.80 (0.69, 0.92)** | **0.003** |
| Rural | 1.26 (0.95, 1.66) | 0.108 | **1.51 (1.14, 2.00)** | **0.004** | **0.75 (0.59, 0.95)** | **0.019** |

Abbreviations: SBS = sexual behavior stigma; CI = confidence interval; OR = Odds ratio.

interval or CI: 1.08, 1.91]), have a 4-year college degree or higher (aOR 1.51 [95% CI: 1.25, 1.81]), and live in rural areas (aOR 1.51 [95% CI: 1.14, 2.00]). Racial minorities (non-Hispanic Black and Hispanic GBM) and GBM with high-school education or less were significantly less likely to be classified in any of the higher stigma categories than in the minimal stigma category.

### Sexual stigma and poverty associations with mental health

Approximately 25% (n = 3094) were missing data on key measures including household size (25%), under-18-year-old dependents (25%), and income (6%). Among participants with available data (n = 9401), the overall prevalence of serious psychological distress was 22% (n = 2058), with higher prevalence among income-poor GBM (35%; 677/1917) compared to income-adequate GBM (18%; 1381/7484). Multivariable regression analyses, using income-adequate GBM identified as experiencing minimal stigma as the reference group, revealed varied associations between sexual stigma and mental distress, with stronger associations observed within income-poor groups (**Table 3** and **Fig 2**). For instance, income-poor GBM identified as experiencing diverse sexual stigma had an adjusted prevalence ratio (aPR) of 4.73 (95% CI: 3.89, 5.74) for serious psychological distress, while their income-adequate counterparts had a slightly lower aPR of 3.89 (95% CI: 3.32, 4.55). Similar trends were noted in the relationships between targeted stigma categories (i.e., family and general social predominant sexual stigma and anticipated healthcare sexual stigma classes) and mental distress, with stronger associations observed in the family and general social stigma groups than in the anticipated healthcare predominant sexual stigma groups.

The prevalence of past-year suicidal ideation and attempts mirrored the patterns observed for mental distress. Overall, suicidal ideation was reported by 20% (1795/9108) of participants, with 29% (535/1822) prevalence among income-poor GBM, and 17% (1260/7286) among income-adequate GBM. The prevalence of suicide attempts was 2% (194/9074) overall, with 5% (91/1807) among income-poor GBM and 1% (103/7267) among income-adequate GBM. Associations between sexual stigma and suicide-related outcomes were consistent with those for mental distress, revealing stronger associations in groups concurrently experiencing poverty. This disparity was most pronounced for the diverse sexual stigma category and suicide attempts, showing an aPR of 11.30 (95% CI: 6.58, 19.41) among income-poor GBM, compared to 3.92 (95% CI: 2.19, 7.00) among income-adequate GBM. In notable contrast to the previously observed pattern, being identified as experiencing predominantly anticipated healthcare sexual stigma showed a slightly stronger association with past-year suicide attempt (aPR 4.26 [95% CI: 2.10, 8.66]) than those in the predominantly family and general social sexual stigma (aPR 3.83 [95% CI: 2.19, 6.69]) among income-poor individuals.

Adjusting for misclassification bias (from uncertainty in stigma class assignments) and selection bias (from missing data) in these associations suggested a bias away from the null (**Table E and Figure B in S2 Text**), with selection bias having a slightly greater impact on the estimated associations than misclassification bias. However, neither significantly altered the directions nor statistical significance of observed associations. Additional results of INR-cutoff sensitivity analysis are shown in **Table G in S2 Text**.

### Interaction quantification

Table 3 also provides the quantified estimates of effect modification by poverty on both multiplicative and additive scales. Most associations exhibited less-than multiplicative interactions, except for the relationships involving the diverse sexual stigma and anticipated healthcare predominant sexual stigma classes and past-year suicide attempts. However, a super-additive (or

**Table 3. Crude and adjusted associations between sexual behavior stigma and mental health outcomes among gay, bisexual and other men who have sex with men, AMIS 2018–2019.**

| Income Poverty Strata | Sexual Behavior Stigma Class (SBSC) | Outcome Prevalence n/N (%) | Prevalence Ratios | | Multiplicative Interaction | Additive Interaction (REPI) |
|---|---|---|---|---|---|---|
| | | | Crude (95% CI) | Adjusted (95% CI) | Estimate (95% CI) | Estimate (95% CI) |
| **Outcome = Serious Psychological Distress (SPD)** | | | | | | |
| Poor | Diverse | 167/290 (58) | 5.87 (4.87, 7.08) | 4.73 (3.89, 5.74) | 0.73 (0.58, 0.94) | 0.27 (-0.38, 1.00) |
| | Anticipated healthcare predom. | 101/253 (40) | 4.07 (3.26, 5.09) | 3.14 (2.49, 3.95) | 0.94 (0.72, 1.25) | 0.48 (-0.14, 1.15) |
| | Family/general social predom. | 279/703 (40) | 4.05 (3.45, 4.75) | 3.32 (2.81, 3.92) | 0.91 (0.74, 1.13) | 0.45 (0.04, 0.89) |
| | Minimal | 130/671 (19) | 1.98 (1.61, 2.42) | 1.69 (1.37, 2.08) | | |
| Not Poor | Diverse | 312/798 (39) | 3.99 (3.41, 4.66) | 3.89 (3.32, 4.55) | | |
| | Anticipated healthcare predom. | 198/966 (20) | 2.09 (1.75, 2.50) | 1.99 (1.66, 2.38) | | |
| | Family/general social predom. | 551/2456 (22) | 2.29 (1.99, 2.63) | 2.17 (1.88, 2.49) | | |
| | Minimal | 320/3264 (10) | *Ref* | *Ref* | | |
| **Outcome = Suicidal Ideation in past year** | | | | | | |
| Poor | Diverse | 130/269 (48) | 4.96 (4.04, 6.08) | 4.00 (3.24, 4.95) | 0.80 (0.62, 1.04) | 0.22 (-0.46, 0.96) |
| | Anticipated healthcare predom. | 75/239 (31) | 3.22 (2.50, 4.14) | 2.52 (1.95, 3.26) | 0.88 (0.65, 1.19) | 0.14 (-0.50, 0.75) |
| | Family/general social predom. | 222/664 (33) | 3.43 (2.89, 4.07) | 2.81 (2.36, 3.36) | 0.94 (0.74, 1.22) | 0.31 (-0.10, 0.75) |
| | Minimal | 108/650 (17) | 1.70 (1.37, 2.12) | 1.46 (1.17, 1.83) | | |
| Not Poor | Diverse | 265/759 (35) | 3.58 (3.04, 4.22) | 3.50 (2.97, 4.13) | | |
| | Anticipated healthcare predom. | 184/929 (20) | 2.03 (1.69, 2.44) | 1.96 (1.63, 2.35) | | |
| | Family/general social predom. | 498/2388 (21) | 2.14 (1.86, 2.46) | 2.04 (1.77, 2.36) | | |
| | Minimal | 313/3210 (10) | *Ref* | *Ref* | | |
| **Outcome = Suicide Attempt in past year** | | | | | | |
| Poor | Diverse | 33/265 (12.5) | 15.96 (9.49, 26.84) | 11.30 (6.58, 19.41) | 1.23 (0.57, 3.13) | 7.63 (3.19, 15.20) |
| | Anticipated healthcare predom. | 12/237 (5.1) | 6.49 (3.26, 12.92) | 4.26 (2.10, 8.66) | 1.39 (0.52, 4.35) | 1.69 (-1.21, 5.21) |
| | Family/general social predom. | 29/657 (4.4) | 5.66 (3.31, 9.66) | 3.83 (2.19, 6.69) | 0.65 (0.31, 1.49) | 0.03 (-2.21, 2.30) |
| | Minimal | 17/648 (2.6) | 3.36 (1.82, 6.23) | 2.47 (1.31, 4.65) | | |
| Not Poor | Diverse | 23/751 (3.1) | 3.93 (2.23, 6.92) | 3.92 (2.19, 7.00) | | |
| | Anticipated healthcare predom. | 9/927 (1.0) | 1.24 (0.58, 2.67) | 1.21 (0.56, 2.61) | | |
| | Family/general social predom. | 46/2384 (1.9) | 2.47 (1.52, 4.03) | 2.34 (1.42, 3.85) | | |
| | Minimal | 25/3205 (0.8) | *Ref* | *Ref* | | |

Abbreviations: CI = confidence interval; REPI = relative excess prevalence due to interaction (quantifies effect modification by income poverty on additive scale for each sexual stigma class compared to the reference category).

Note: All adjusted models were adjusted for participant age, race/ethnicity, nativity, US region of residence and urbanicity.

greater-than additive) interaction was observed for all associations, indicating that the co-occurrence of patterns of sexual stigma experiences exceeding minimal levels and poverty was associated with excess prevalence of all three mental health outcomes beyond the expected sum of prevalences when either condition occurred alone. This additive effect was statistically significant for the association between the diverse sexual stigma class and past-year suicide attempts.

## Discussion

This study examined sexual stigma experiences among HIV-negative GBM, identifying four distinct groups with nuanced vulnerabilities across specific GBM demographics. These groups

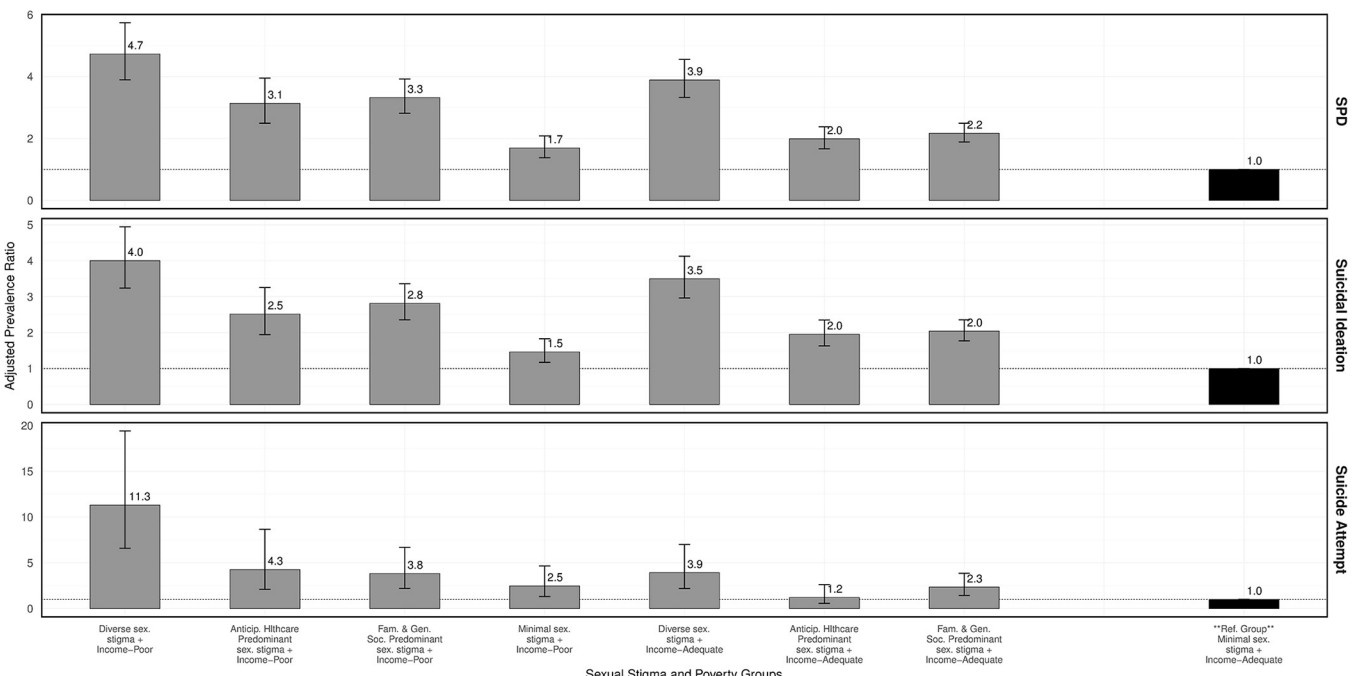

**Fig 2. Adjusted prevalence ratios of sexual stigma association with mental health outcomes among US GBM and variations by poverty, AMIS 2018–2019.**
Error bars represent 95% confidence interval; SPD = serious psychological disorder.

showed varying associations with mental distress and suicide-related outcomes, while highlighting the amplifying impact of economic hardship on the relationship between sexual stigma and GBM mental health. These findings suggest that addressing stigma in the context of structural determinants like poverty may be key to optimizing mental health among GBM.

The four identified stigma classes illustrate the diversity of sexual stigma stressors faced by GBM. Drawing on both MST and PMF, these categories shed light on potential maladaptive coping strategies that may emerge in response to these stressors [11,12]. For instance, individuals in in the "Anticipated Healthcare Predominant Sexual Stigma" class, anticipating sexual behavior-related discrimination within healthcare settings, may adopt avoidant coping strategies that hinder their access to necessary care. In contrast, those in the "Diverse Sexual Stigma" class, who have encountered a wider range of stigma-related stressors, may develop a broader array of maladaptive coping mechanisms including social withdrawal, sexual behavior disclosure avoidance, compartmentalization, as well as avoidant coping [11,12]. These align with previous research linking distinct stigma experiences to specific maladaptive coping response; anticipated stigma is associated with avoidant coping, while perceived stigma is linked to adoption of distancing behaviors and engagement in attack/escape avoidance coping strategies [66,67]. Understanding these stigma patterns can inform targeted interventions that promote adaptive coping and mitigate the associated psychological consequences among GBM.

Vulnerabilities to distinct sexual stigma groups varied by key demographics, with younger age individuals more likely to be in the higher-than-minimal stigma classes, consistent with prior research [68]. Despite ongoing stigma against sexual minorities, improvements in legal protections, media visibility, and leadership representation in the US and other developed countries suggest a more favorable social environment for younger GBM than in the past [4,5,69]. While these improvements remain insufficient, they may encourage greater openness about sexual orientation among younger GBM, potentially increasing their exposure to sexual

stigma. In contrast, older GBM, having lived through less accepting times, may exhibit lower self-disclosure. This age-related vulnerability towards higher-than-minimal sexual stigma categories observed in this study may also be a reflection of a heightened sexual stigma awareness among younger GBM, making them more likely to recognize and report stigma experiences compared to older GBM. Other notable associations include foreign-born status and high education being linked to a greater likelihood of being in the Anticipated Healthcare Predominant Sexual Stigma class, potentially reflecting vulnerabilities shaped by prior healthcare discrimination (foreign-born GBM) and heightened awareness of heterosexism (educated GBM) [70,71]. Interestingly, racial minorities were significantly less likely to belong to these higher-than-minimal stigma classes, raising questions about how cultural identity influences the perception and reporting of sexual stigma. Collectively, these findings highlight the intersectional forces that shape sexual stigma perception and reporting, emphasizing the need for adaptive interventions that consider these complexities.

The prevalence of serious psychological distress among GBM in this study was notably higher than the 11% reported in an analysis of the 2017 National HIV Behavioral Survey data for Tennessee, which also utilized the K-6 scale with a similar cutoff [72]. Population-based surveys from the same period, employing the same mental distress scale cut-off, reported distress prevalences of around 4%, highlighting the substantial disparity between GBM and the wider US adult population [73,74]. These findings emphasize the urgent need to address the unique mental health challenges faced by GBM. In conjunction with prior MST-based research that highlighted the buffering role of social support on mental health problems, this study's finding that those with predominantly family and general social sexual stigma experiences had a slightly stronger association with serious psychological distress and suicidal ideation than those with predominantly anticipated healthcare sexual stigma highlights the need to recognize and address stigma from these sources in mental health interventions [12,75]. While pharmaceutical approach is crucial in mental healthcare, a holistic strategy that also tackles the traumas linked to specific stigma patterns may offer a more effective mental health support that surpasses the benefits of solely focusing on clinical treatment.

This study highlighted the intensified impact of sexual stigma on mental health in the presence of economic hardship. We hypothesize that the stronger association between the Anticipated Healthcare Predominant Sexual Stigma class and suicide attempts among economically disadvantaged GBM, may stem from the combined effects of avoidant coping and reduced healthcare affordability. These barriers hinder effective healthcare engagement, elevating the likelihood of suicide attempts in those experiencing mental distress compared to their peers without these dual challenges. Acknowledging the importance of additive effect measure modification in identifying target groups for public health interventions, our findings suggest that prioritizing mental health interventions for income-poor GBM who may be underserved by existing resources to avoid perpetuating disparities in mental health outcomes [61,76].

This study had several limitations. While we used a diverse sample of GBM, it may not fully represent specific GBM demographics, particularly racial minorities, socioeconomically disadvantaged individuals, and HIV-status unaware GBM, affecting generalizability of the study findings. Reliance on self-reported data introduces potential biases like recall and social desirability, which were not accounted for in the analysis. However, if present, these biases may skew the results towards the null, indicating a potential underestimation or attenuation of true associations. Additionally, using the same tool to measure both exposure and outcome introduces the possibility of dependent misclassification error, worsening biases. Finally, the cross-sectional study design limits causal inference regarding explored relationships. Additionally, the focus on individual-level sexual stigma overlooks other dimensions of sexual stigma faced by GBM, including institutional, and community-level sexual stigma, as well as other stigma

forms (mental health, HIV-related, etc.), along with their intersections with other systemic forces like racism that also affect mental health among GBM.

## Public health implications and future directions

This study emphasizes the necessity for comprehensive approaches that encompass addressing socio-structural factors when addressing mental well-being among HIV-uninfected GBM. It is important that interventions prioritize reducing sexual stigma stressor events across all settings, including healthcare, through mitigation campaigns, staff education, and legal protections against non-heteronormative discrimination. Addressing the compounding role of poverty is also crucial, and policies offering free or highly subsidized mental health services for qualifying HIV-negative GBM may enhance access for those with higher-than-minimal sexual stigma experiences. In healthcare settings, training clinical staff to recognize demographic vulnerabilities and deliver culturally sensitive services may reduce anticipated healthcare stigma and mitigate its adverse impacts on mental health. Promising directions for future research include longitudinal studies to help understand links between distinct stigma patterns, coping strategies, and social support degradation, as well as assess the effectiveness of tailoring mental health support to distinct patterns of stigma exposure, particularly within the context of poverty.

## Supporting information

**S1 Text. Appendix.**
(DOCX)

**S2 Text. Supplemental tables and figures.**
(DOCX)

## Acknowledgments

We extend our sincere appreciation to our AMIS participants for their invaluable contributions and willingness to share their experiences and insights, without which this study would not have been possible. We also acknowledge the valuable support of Drs. Lash and Naimi of the Department of Epidemiology, Emory university for their mentorship and support with various aspects of this study.

## Author Contributions

**Conceptualization:** Udodirim N. Onwubiko, Allison T. Chamberlain, David Benkeser, David P. Holland, Samuel M. Jenness, Stefan D. Baral.

**Data curation:** Udodirim N. Onwubiko, Travis H. Sanchez, Samuel M. Jenness, Stefan D. Baral.

**Formal analysis:** Udodirim N. Onwubiko, Sarah M. Murray, Amrita Rao, David Benkeser, Samuel M. Jenness, Stefan D. Baral.

**Funding acquisition:** Travis H. Sanchez, Samuel M. Jenness, Stefan D. Baral.

**Methodology:** Sarah M. Murray, Amrita Rao, David Benkeser, Stefan D. Baral.

**Project administration:** Stefan D. Baral.

**Supervision:** Allison T. Chamberlain, David P. Holland, Samuel M. Jenness, Stefan D. Baral.

**Writing – original draft:** Udodirim N. Onwubiko.

**Writing – review & editing:** Udodirim N. Onwubiko, Sarah M. Murray, Amrita Rao, Allison T. Chamberlain, Travis H. Sanchez, David Benkeser, David P. Holland, Samuel M. Jenness, Stefan D. Baral.

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
