## [Decision Letter · Decision Letter 0]

9 Sep 2024

PMEN-D-24-00243

Intersecting Realities: Understanding Stigma, Poverty, and Mental Health in HIV-Negative Men who have Sex with Men in the United States

PLOS Mental Health

Dear Dr. Onwubiko,

Thank you for submitting your manuscript to PLOS Mental Health and we apologise for the delay in reaching a decision - thank you for your patience and understanding. After careful consideration, and having seen the reviewer comments, we feel that it has merit but does not fully meet PLOS Mental Health’s publication criteria as it currently stands. Therefore, we invite you to submit a revised version of the manuscript that addresses the points raised during the review process.

Please ensure that you address all of the comments raised by the Reviewers, which you will be able to see in full at the end of this email. 

We look forward to receiving your revised manuscript.

Kind regards,

Karli Montague-Cardoso

Executive Editor

PLOS Mental Health

Journal Requirements:

https://journals.plos.org/mentalhealth/s/figures 

https://journals.plos.org/mentalhealth/s/figures#loc-file-requirements 

Additional Editor Comments (if provided):

Reviewers' comments:

Reviewer's Responses to Questions

**Comments to the Author**

1. Does this manuscript meet PLOS Mental Health’s publication criteria? Is the manuscript technically sound, and do the data support the conclusions? The manuscript must describe methodologically and ethically rigorous research with conclusions that are appropriately drawn based on the data presented.

Reviewer #1: Yes

Reviewer #2: Yes

2. Has the statistical analysis been performed appropriately and rigorously?

Reviewer #1: No

Reviewer #2: Yes

3. Have the authors made all data underlying the findings in their manuscript fully available (please refer to the Data Availability Statement at the start of the manuscript PDF file)?

Reviewer #1: Yes

Reviewer #2: No

4. Is the manuscript presented in an intelligible fashion and written in standard English?

Reviewer #1: Yes

Reviewer #2: Yes

5. Review Comments to the Author

Reviewer #1: Dear authors

I have reviewed the manuscript titled "Intersecting Realities: Understanding Stigma, Poverty, and Mental Health in HIV-Negative Men who have Sex with Men in the United States." The study explores the intersection of sexual stigma, poverty, and mental health among HIV-negative men who have sex with men (MSM) in the United States using data from the 2018 and 2019 American Men's Internet Survey (AMIS).

Without a doubt, this study addresses highly relevant social and public health issues, namely sexual stigma, poverty, and mental health in men who have sex with men (MSM). These factors are critical because MSM historically face high levels of discrimination and stigmatization, contributing to significant disparities in mental health and access to healthcare services. However, I have some concerns, which I will comment on below:

The manuscript provides a strong rationale for the study. However, it would be beneficial to focus more on the specific gap that this study addresses within the current body of literature, especially regarding HIV-negative MSM. For instance, the authors could discuss existing studies' limitations to underscore this work's significance.

Regarding the analytical strategy, while I appreciate the use of latent class analysis (LCA) to capture heterogeneity in the experiences of sexual stigma among MSM, I have some reservations. My main concern is about the use of a combined dataset, as I believe it might be impossible to analyze the overlap between the 2018 and 2019 datasets. This problem raises the question of to what extent this overlap might bias the study's results. Specifically, there is a risk of overrepresenting certain groups, which could, in turn, skew the analysis of the relationship between the latent classes and other variables. Given the substantial sample sizes of both datasets, combining them seems unnecessary and wrong. I suggest using only one dataset or separately presenting the results for both.

Additionally, I have questions regarding the use of multinomial logistic regression in the analysis. It is unclear whether dummy variables were used for the categorical variables, which is the recommended approach. Using dummy variables ensures that the categories are appropriately represented and compared in the regression models. Clarifying this aspect of the methodology would enhance the robustness of the analysis.

Moreover, the authors state: "Membership in diverse-SBS and FGSP-SBS classes were associated with younger age, past-year homelessness, poverty, and residence in the US South or Midwest" (page 9, line 182). However, based on the reported data, this is only true when compared to the reference class (which should be indicated in the table). Specifying the reference class used in the comparisons is crucial for accurately interpreting these associations.

Finally, the discussion should be revised based on the results obtained after the requested changes. This will ensure that the interpretation and implications of the findings are aligned with the revised analysis, providing a more accurate and robust conclusion.

Minor Issues

Page 1, lines 9, 12, and 16: check citation format.

Page 9, line 177: the title of a figure is inserted.

Ensure that all software used for the analyses is properly cited and referenced.

Reviewer #2: Thank you for the invitation to review this quantitative study examining the intersection of MSM stigma and poverty among MSM in the US. The analyses are straightforward, and the results have the potential to influence public health policies and interventions for this underserved group. My main suggestion for the authors is to include more clarification around terminologies, methods, and analyses to make the manuscript more accessible to a wider audience. Many new terms are introduced in the latter part of the manuscript, and these should have been defined in the introduction section.

Line 3: Can clarification be provided on whether the authors are only referring to interpersonal stigma, or if they also include systemic, structural, and internal forms?

Page 1: While I appreciate the general description of stigma and its link to minority stress and psychological mediation, the concept of stigma should be further nuanced by addressing the specific injustices that MSM face.

Line 29: ad – typo?

Line 32: Some justification is needed for prioritising 'socioeconomic disparity' over other systemic forces like racism in this paper. I also recommend revising the title to more accurately reflect the paper's objective, as 'Intersecting Realities' is too broad for its scope.

Line 39: Can I recommend citing Carmen Logie's work on identifying sexual orientation (including identity, behaviour, etc.) as a determinant of health, alongside other crucial factors such as socioeconomic status, to strengthen your case for examining these multiplicative determinants of health? https://doi.org/10.2105/AJPH.2011.300599

Page 56: Add a sentence about the recruitment strategy.

Line 70: A sentence is needed regarding the validity and reliability of the K6 measure for this sample.

Line 78: A sentence explaining how these sexual stigma items were designed or chosen needs to be added.

Line 85: A brief definition is needed in the introduction section about enact and anticipated stigma.

Line 88: It will be acceptable to refer to 'sexual orientation' here only if the introduction includes a definition that broadly encompasses sexual identity, behaviour, preference, etc.

Line 89: More clarification is needed about the categorisation of stigma in different settings. Not quite sure what line 89 means. Can the language be simplified or more elaboration be added?

Page 6: Further clarification (in layman terms) is needed on how the categories are determined: diverse forms of stigma across multiple settings; primarily anticipated stigma in healthcare settings; predominantly enacted and perceived sexual stigma in family and general social settings; or minimal sexual stigma.

Line 227: Definitions are needed for 'multiplicative' and 'additive'.

6. PLOS authors have the option to publish the peer review history of their article (what does this mean?). If published, this will include your full peer review and any attached files.

**Do you want your identity to be public for this peer review?** For information about this choice, including consent withdrawal, please see our Privacy Policy.

Reviewer #1: **Yes: **Fabiola Gómez

Reviewer #2: No

---

## [Decision Letter · Decision Letter 1]

6 Dec 2024

Exploring the Intersections of Sexual Stigma, Poverty and Mental Health in HIV-Negative Gay, Bisexual and Other Men Who Have Sex with Men in the United States

PMEN-D-24-00243R1

Dear Dr. Onwubiko,

We are pleased to inform you that your manuscript 'Exploring the Intersections of Sexual Stigma, Poverty and Mental Health in HIV-Negative Gay, Bisexual and Other Men Who Have Sex with Men in the United States' has been provisionally accepted for publication in PLOS Mental Health.

Best regards,

Karli Montague-Cardoso

Staff Editor

PLOS Mental Health

Reviewer Comments (if any, and for reference):

Reviewer's Responses to Questions

**Comments to the Author**

1. If the authors have adequately addressed your comments raised in a previous round of review and you feel that this manuscript is now acceptable for publication, you may indicate that here to bypass the “Comments to the Author” section, enter your conflict of interest statement in the “Confidential to Editor” section, and submit your "Accept" recommendation.

Reviewer #1: All comments have been addressed

Reviewer #2: All comments have been addressed

2. Does this manuscript meet PLOS Mental Health’s publication criteria? Is the manuscript technically sound, and do the data support the conclusions? The manuscript must describe methodologically and ethically rigorous research with conclusions that are appropriately drawn based on the data presented.

Reviewer #1: Yes

Reviewer #2: Yes

3. Has the statistical analysis been performed appropriately and rigorously?

Reviewer #1: Yes

Reviewer #2: Yes

4. Have the authors made all data underlying the findings in their manuscript fully available (please refer to the Data Availability Statement at the start of the manuscript PDF file)?

Reviewer #1: Yes

Reviewer #2: No

5. Is the manuscript presented in an intelligible fashion and written in standard English?

Reviewer #1: Yes

Reviewer #2: Yes

6. Review Comments to the Author

Reviewer #1: I have thoroughly reviewed the revised manuscript titled Exploring the Intersections of Sexual Stigma, Poverty, and Mental Health in HIV-Negative Gay, Bisexual and Other Men Who Have Sex with Men in the United States as well as the authors’ responses and clarifications. I sincerely appreciate the attention and effort dedicated to addressing the comments raised during the previous review.

I believe the revisions have significantly enhanced the quality of the manuscript, both theoretically and methodologically. The integration of concepts related to sexual stigma and poverty, along with their implications for mental health, provides a robust and relevant analysis, supported by a rigorous methodological framework.

I have no further comments or suggestions. I commend the authors for their work, and I am confident this article represents a valuable contribution to the field.

Kind regards,

Fabiola

Reviewer #2: Thank you for thoroughly considering the suggestions and providing thoughtful responses. I have no further comments and look forward to seeing the article published.

7. PLOS authors have the option to publish the peer review history of their article (what does this mean?). If published, this will include your full peer review and any attached files.

**Do you want your identity to be public for this peer review?** For information about this choice, including consent withdrawal, please see our Privacy Policy.

Reviewer #1: **Yes: **Fabiola Gómez

Reviewer #2: **Yes: **Kyle Tan
